# Semantic Image Synthesis with Semantically Coupled VQ-Model

**Stephan Alaniz**[*†‡]      **Thomas Hummel**[*†]      **Zeynep Akata**[†‡]

[†]Cluster of Excellence Machine Learning, University of Tübingen
[‡]Max Planck Institute for Informatics
[*]Equal contribution
firstname.lastname@uni-tuebingen.de

## Abstract

Semantic image synthesis enables control over unconditional image generation by allowing guidance on what is being generated. We conditionally synthesize the latent space from a vector quantized model (VQ-model) pre-trained to autoencode images. Instead of training an autoregressive Transformer on separately learned conditioning latents and image latents, we find that jointly learning the conditioning and image latents significantly improves the modeling capabilities of the Transformer model. While our jointly trained VQ-model achieves a similar reconstruction performance to a vanilla VQ-model for both semantic and image latents, tying the two modalities at the autoencoding stage proves to be an important ingredient to improve autoregressive modeling performance. We show that our model improves semantic image synthesis using autoregressive models on popular semantic image datasets ADE20k, Cityscapes and COCO-Stuff.

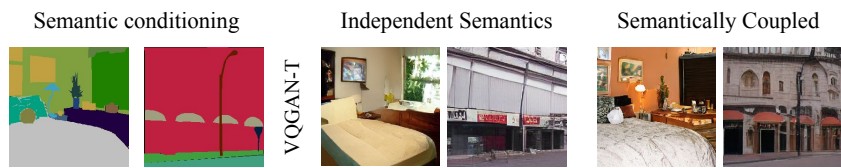

Figure 1: A semantically coupled VQ-model together with a Transformer generator synthesizes images that follows the semantic guidance closer and has higher fidelity. For instance, the semantically coupled model correctly reproduces the lamp next to the bed and more accurately matches the shape of the store fronts.

## 1 Introduction

Semantic image synthesis allows for precise specification of the semantic content of an image. This enables applications such as artistic image creation, e.g. by outlining the scene and components in novel ways, or data augmentation (Shetty et al., 2020), e.g. creating similar images or changing objects or styles. In this work, semantic information refers to the class identity of objects (e.g. person, car, dog, chair) and scene concepts (e.g. sky, road, grass, lake), but also their locations, size and shape in the image. Advances in generative models have led the progress in semantic image synthesis methods mostly through improvements to GAN-based models (Isola et al., 2017; Hong et al., 2018; Park et al., 2019; Ntavelis et al., 2020) and autoregressive generative models (Chen et al., 2020; Child et al., 2019; Razavi et al., 2019). Vector-quantized models (VQ-model) such as the VQGAN model (Esser et al., 2021) combine the benefits of both GANs and autoregressive training into a single model. By building upon the VQVAE (van den Oord et al., 2017; Razavi et al., 2019), the addition of a discriminator results in high-fidelity image generations similar to other GAN-based models.

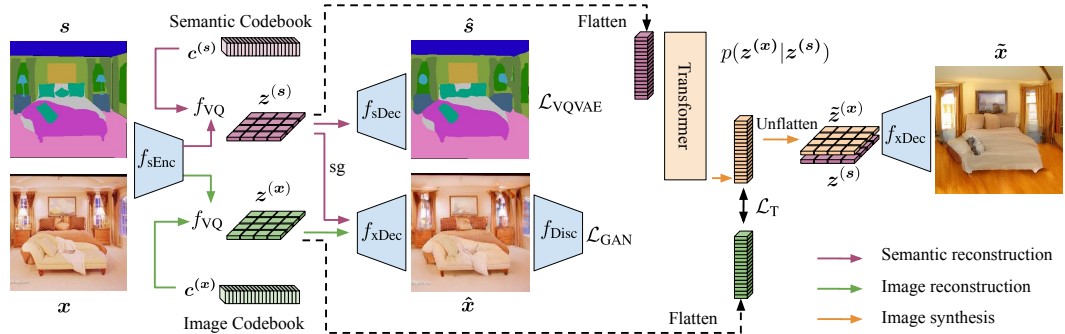

Figure 2: Overview of our semantically coupled VQ-model architecture. A single encoder produces both semantic and image latents. Two decoders reproduce semantics and image, while the image decoder also uses semantic information. The Transformer is trained to predict image latents conditioned on semantic latents and the image decoder synthesizes the latents.

In this work we propose improvements to the architecture of VQ-models, such as the VQGAN, that allows more effective usage of semantic conditioning information. Our model incorporates semantic information already at the autoencoding stage, allowing the VQ-model to combine the two modalities before the autoregressive training stage. We train a Transformer model (Vaswani et al., 2017) to generate latents which can subsequently be synthesized by the decoder of the VQ-model.

To accomplish this, we make the following contributions. We propose an extension to the VQ-model auto-encoder that incorporates the semantic information used for image synthesis. By doing so, the decoder already learns the relationship between semantics and image content to make better image generations. An autoregressive Transformer model then only needs to act on the latent space of a single encoder model as opposed to requiring an auxiliary VQVAE which was proposed by Esser et al. (2021). Our semantically coupled latents enable the Transformer model to create a better generative model of the data as seen in Figure 1, where semantic detail is better replicated in the synthesized images.

## 2 MODEL

We follow the methodology of Esser et al. (2021) to produce a generative model with semantic conditioning as a two-step approach. Instead of directly generating on the image space, we perform the task on the latent space. To do so, our image synthesis pipeline consists of two models parts: 1) an auto-encoder that learns a latent representation of the images; 2) an autoregressive model that learns to generate the latent code.

**Auto-Encoding with VQ-Models.** We consider both VQVAE (van den Oord et al., 2017; Razavi et al., 2019) and VQGAN (Esser et al., 2021) as the latent variable models and refer to them collectively as VQ-models. A VQ-model consists of an encoder $f_{\text{Enc}}$ that maps the input $x$ to a discrete latent space $z$ and a decoder $f_{\text{Dec}}$ reconstructing $x$ from $z$. The output of the encoder $f_{\text{Enc}}(x)$ is quantized to the closest vector of a learned codebook $c \in \mathbb{R}^{K \times D}$ where $K$ is the number of codebook entries and $D$ the dimensionality. The quantization of the latent space allows using autoregressive generative models on shorter sequences than the full input data dimension. VQGAN sets itself apart from the VQVAE by introducing an additional discriminator CNN that is trained to distinguish between ground truth images and reconstructions from the decoder $f_{\text{Dec}}$.

**Autoregressive Modeling with Transformer.** After training the VQ-model on images $x$ completes, an autoregressive model is typically trained to generate the image latents $z^{(x)}$ of the training images, by maximizing the likelihood of the factorized joint distribution

$$p(z^{(x)}) = p(z_1^{(x)}, ... z_n^{(x)}) = \prod_i^n p(z_i^{(x)} | z_1^{(x)}, \ldots, z_{i-1}^{(x)}). \tag{1}$$

| | Cityscapes | | | ADE20k | | | COCO-Stuff | | |
|---|---|---|---|---|---|---|---|---|---|
| | FID↓ | SSIM↑ | LPIPS↓ | FID↓ | SSIM↑ | LPIPS↓ | FID↓ | SSIM↑ | LPIPS↓ |
| VQVAE-T (van den Oord et al., 2017; Razavi et al., 2019) | 190.22 | 0.3652 | 0.6406 | 142.11 | 0.0769 | 0.8043 | 111.85 | 0.0820 | 0.8127 |
| sVQVAE-T | 192.65 | **0.4000** | 0.6002 | 148.09 | **0.1343** | 0.7846 | 114.55 | **0.1382** | 0.7857 |
| VQGAN-T (Esser et al., 2021) | 130.49 | 0.2797 | 0.4873 | 46.50 | 0.0667 | 0.6460 | 33.38 | 0.0638 | 0.6533 |
| sVQGAN-T | 131.37 | 0.3013 | **0.4034** | **38.36** | 0.0987 | **0.5534** | **28.80** | 0.0984 | **0.5583** |

Table 1: Semantic image synthesis results for VQ-Tansformer models on Cityscapes, ADE20K and COCO-Stuff datasets measuring SSIM, FID and LPIPS between generations and ground truth images with the same semantic map. (sVQVAE-T and sVQGAN-T uses $\lambda = 0.1$)

In order to perform semantic image synthesis, Esser et al. (2021) trains a separate VQVAE on auto-encoding the semantic map $s$ to produce semantic latents $z^{(s)}$. To condition the autoregressive model with this semantic information, we prepend the semantics latents $z^{(s)}$ to the image latents $z^{(x)}$ and train the autoregressive model on the conditional likelihood

$$p(z^{(x)}|z^{(s)}) = \prod_i^n p(z_i^{(x)}|z_1^{(x)}, \ldots, z_{i-1}^{(x)}, z^{(s)}) \qquad (2)$$

which is done by minimizing the negative log-likelihood $\mathcal{L}_T = -\log p(z^{(x)}|z^{(s)})$. For the autoregressive model, we use a Transformer (Vaswani et al., 2017).

**Semantically Coupled VQ-Model.** We deem the existing approach of conditioning semantic information to the autoregressive Transformer suboptimal for several reasons. It requires training two independent VQ-models and the decoder only uses the image latents to produce the reconstruction, while more information in form of the semantic latents is available. This shifts the learning of correlations and dependencies between semantics and image entirely to the Transformer. We propose a semantically coupled VQ-model that incorporates the conditioning information already in the auto-encoding stage of a single VQ-model that jointly learns to reconstruct both images and semantics.

Figure 2 illustrates the joint model learning both latents at the same time. The encoder $f_{\text{sEnc}}(x, s)$ is a shared encoder that takes the concatenation of the image and semantic map as input to produce two latents, $z^{(x)}$, and $z^{(s)}$, respectively. The decoder is then split into two CNNs, one reconstructing the semantics using only the semantic latent $f_{\text{sDec}}(z^{(s)}) = \hat{s}$ and one reconstructing the image having access to both the semantic and the image latent $f_{\text{xDec}}(z^{(x)}, \text{sg}[z^{(s)}]) = \hat{x}$. We stop the gradient flow from the image decoder to the semantic latent, such that each decoder is responsible for training exactly one of the latents, separating and focusing their training signal, while still allowing the image encoder to access the semantic latent and, thus, allowing it to encode complementary information in the image latents.

By restricting the gradient flow and using two decoders, we also induce the dependency structure of the two latents, i.e., the image latent depends on the semantic latent, but not vice versa, and the semantic latent is learned independently. Apart from this architectural change, the loss functions remain the same for both VQVAE and VQGAN. The semantic reconstruction is again trained with a cross-entropy term. Thus, the loss terms are combined into a single loss function

$$\mathcal{L}_{\text{sVQ}} = \mathcal{L}_{\text{VQ}}(x, f_{\text{xDec}}(z^{(x)}, z^{(s)})) + \lambda \mathcal{L}_{\text{VQVAE}}(s, f_{\text{sDec}}(z^{(s)})) \qquad (3)$$

where the second term comes from the VQVAE reconstructing the semantics and $\mathcal{L}_{\text{VQ}}$ concerns reconstructing the image and can be either $\mathcal{L}_{\text{VQGAN}}$ or $\mathcal{L}_{\text{VQVAE}}$. The hyperparameter $\lambda$ allows balancing the two loss terms.

After training the semantic VQ-model to auto-encode both images and semantics, we train the Transformer network with a cross-entropy loss to maximize the log-likelihood of Equation 2 where both the conditioning latents and the prediction latents come from the same VQ-model.

## 3 EXPERIMENTS

**Experimental Setup.** To train and evaluate our models, we combine several semantic image datasets into one large dataset with dense semantic image annotations, namely Cityscapes (Cordts et al.,

Semantics    VQGAN-T     sVQGAN-T

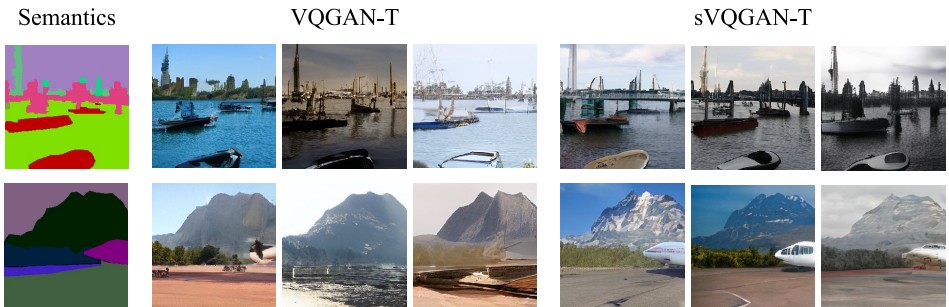

Figure 3: Qualitative results comparing multiple generations of an individually trained VQGAN-T with a semantically coupled sVQGAN-T on COCO-Stuff (top) and ADE20K (bottom). Semantic details are only retained by sVQGAN-T, e.g., bridge (top), airplane (bottom).

2016), ADE20k (Zhou et al., 2017) and COCO-Stuff (Caesar et al., 2018). We create a unified semantic class mapping across the three datasets combining the 20, 150 and 183 object classes of Cityscapes, ADE20k and COCO-Stuff into a total of 243 classes by merging labels that occur across datasets such as person, car, building, etc. We evaluate our semantically coupled VQ-model in comparison to the traditional approach of training semantic latents and image latents separately. We use the VQVAE and VQGAN as base models and evaluate the VQ-models after the second stage when performing semantic image synthesis. The quality of image reconstructions and generations is evaluated using the Fréchet Inception Distance (FID) (Heusel et al., 2017), the structural similarity index (SSIM) (Wang et al., 2004), and the Learned Perceptual Image Patch Similarity (LPIPS) (Zhang et al., 2018).

**Semantic Image Synthesis using VQ-Transformer.** We find that semantically coupling the latents with our sVQVAE or sVQGAN model significantly improves the performance of the Transformer model in predicting the conditional sequence of latents. In Table 1, we observe that our semantically coupled VQ-model improves over the individually trained models across all datasets and metrics. For instance, the FID score of sVQGAN-T significantly improves over VQGAN-T with 38.4 vs. 46.5 on ADE20k and 28.8 vs. 33.4 on COCO-Stuff (lower is better). Thus, sVQGAN-T can better model the whole data distribution of the original datasets, which includes covering all semantic classes without distortions. For both VQVAE-T and VQGAN-T models, we find that our semantic variants achieve better SSIM and LPIPS scores, e.g., LPIPS of sVQVAE-T is 0.403 vs 0.487; 0.553 vs 0.646; 0.558 vs 0.653 (lower is better). These results indicate that our semantically coupled VQ-models better follow the semantic structure of the image as the semantic maps are the only source of information about the ground truth image which the metrics use for evaluation. We find that the improvements of our semantically coupled VQ-models stem from the complementary structure the semantic and image latents have learned during the auto-encoding training stage. In Figure 5, we illustrate synthesized images from the VQGAN-T and the sVQGAN-T model, sampling three times each. Some details of the semantic information is sometimes not properly replicated by the VQGAN-T model, e.g., the bridge in the first row or the plane in the second row. On the other hand, our sVQGAN-T model consistently generates these details provided by the semantics. These results show that training the latents of the two modalities together create stronger dependencies between them, which the Transformer model can leverage.

## 4   CONCLUSION

In this work, we present semantically coupled VQ-models that jointly learn latents of two modalities, images and semantic maps. We have shown that coupling the latents during training leads to dependencies that are easier to pick up by the Transformer model used to model their conditional distribution. For both VQVAE and VQGAN as the VQ-model, the semantic coupling improves the synthesis of images especially in following details of the semantic maps that is being conditioned on. Further investigation into understanding the cause of our findings could allow designing latent variable models with better synergies across data modalities beneficial for autoregressive modeling of Transformers. Currently, a reference image is required during inference as input to the VQ-Model. Further work will try to alleviate this dependency completely.

## ACKNOWLEDGMENTS

The authors would like to thank Scott Reed for valuable discussions and insightful feedback. This work has been partially funded by the ERC (853489 - DEXIM) and by the DFG (2064/1 – Project number 390727645). The authors thank the International Max Planck Research School for Intelligent Systems (IMPRS-IS) for supporting Thomas Hummel.

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

# A    AUTO-ENCODING WITH VQ-MODELS

We also inspect how the architectural changes of our semantically coupled VQ-model influence the first stage of training, i.e., auto-encoding both semantic and image content. Regardless of the VQ-model, we use the same architecture for all VQVAE and VQGAN models respectively. All VQ-models use a codebook size of 16384 for the image latent and 4096 for the semantic latent. The latent size is $16 \times 16$ for both the semantic and the image latent.

## A.1    AUTO-ENCODING OF IMAGES WITH VQ-MODELS

In Table 2, we report the reconstruction results on images evaluating both VQVAE and VQGAN models on Cityscapes, ADE20k and COCO-Stuff after being trained on our unified dataset. We find that the individual models tend to get better FID, SSIM and LPIPS scores than our semantically coupled variants, e.g., SSIM of VQVAE is better than sVQVAE (0.648 vs 0.620; 0.427 vs 0.397; 0.408 vs 0.382). This indicates that focusing the auto-encoding on image only obtains better image reconstructions when this is the only application. The gap is smaller for VQGAN models, where SSIM of sVQGAN with $\lambda = 0.1$ is close (0.584 vs 0.571; 0.368 vs 0.355; 0.343 vs 0.332). While FID and LPIPS metrics are also close in general, our sVQGAN with $\lambda = 0.1$ obtains a better FID score for ADE20k (45.01 vs 45.57) and COCO-Stuff (34.79 vs 34.93) than the individually trained VQGAN. A better FID score can be achieved, as the image decoder obtains not only the image latents, but also the semantic latents, thus, allowing the decoder to access additional information. Since SSIM and LPIPS favor reconstructions that contains as much information about the ground truth image as possible, the semantic side information is not improving those metrics. As soon as we consider both modalities in the second training stage, the combination will become essential.

| | Cityscapes | | | ADE20k | | | COCO-Stuff | | |
|---|---|---|---|---|---|---|---|---|---|
| | FID↓ | SSIM↑ | LPIPS↓ | FID↓ | SSIM↑ | LPIPS↓ | FID↓ | SSIM↑ | LPIPS↓ |
| VQVAE (van den Oord et al., 2017; Razavi et al., 2019) | 125.17 | **0.6484** | 0.3921 | 67.44 | **0.4267** | 0.5483 | 49.14 | **0.4081** | 0.5583 |
| sVQVAE ($\lambda = 0.1$) | 128.76 | 0.6199 | 0.4504 | 70.03 | 0.3966 | 0.6123 | 50.57 | 0.3820 | 0.6160 |
| VQGAN (Esser et al., 2021) | **123.76** | 0.5844 | **0.1385** | 45.57 | 0.3683 | **0.1905** | 34.93 | 0.3429 | **0.1874** |
| sVQGAN ($\lambda = 0.01$) | 124.61 | 0.5544 | 0.1547 | 45.97 | 0.3444 | 0.2081 | 35.19 | 0.3217 | 0.2046 |
| sVQGAN ($\lambda = 0.1$) | 124.10 | 0.5709 | 0.1506 | **45.01** | 0.3554 | 0.2053 | **34.79** | 0.3318 | 0.2019 |
| sVQGAN ($\lambda = 1.0$) | 125.48 | 0.5444 | 0.1682 | 46.70 | 0.3316 | 0.2311 | 35.60 | 0.3138 | 0.2252 |

Table 2: Auto-encoding results for VQ-models on Cityscapes, ADE20K and COCO-Stuff datasets. Structural similarity (SSIM), Fréchet inception distance (FID) and LPIPS between reconstructions and ground truth images.

## A.2    AUTO-ENCODING OF SEMANTIC INFORMATION WITH VQ-MODELS

In Table 3, we report the mean intersection over union (mIOU) of a VQVAE trained on reconstructing the semantics alone compared to the semantic reconstruction of our semantically coupled models. When only assessing VQVAE and sVQVAE, we observe a similar picture as with the image reconstructions. The quantitative results suggest training individual models is preferable if

| | Cityscapes | ADE20k | COCO-Stuff |
|---|---|---|---|
| VQVAE (van den Oord et al., 2017; Razavi et al., 2019) | 85.45 | 83.84 | 92.38 |
| sVQVAE ($\lambda = 0.1$) | 68.13 | 47.75 | 80.15 |
| sVQGAN ($\lambda = 0.01$) | 80.36 | 79.09 | 87.52 |
| sVQGAN ($\lambda = 0.1$) | 85.90 | 88.77 | 93.40 |
| sVQGAN ($\lambda = 1.0$) | **86.81** | **90.37** | **93.81** |

Table 3: Auto-encoding semantic results for VQ-models on Cityscapes, ADE20k and COCO-Stuff. Mean intersection over union (mIOU) in percent between reconstructions and ground truth semantics, higher is better.

GT      VQVAE*      sVQVAE      VQGAN      sVQGAN

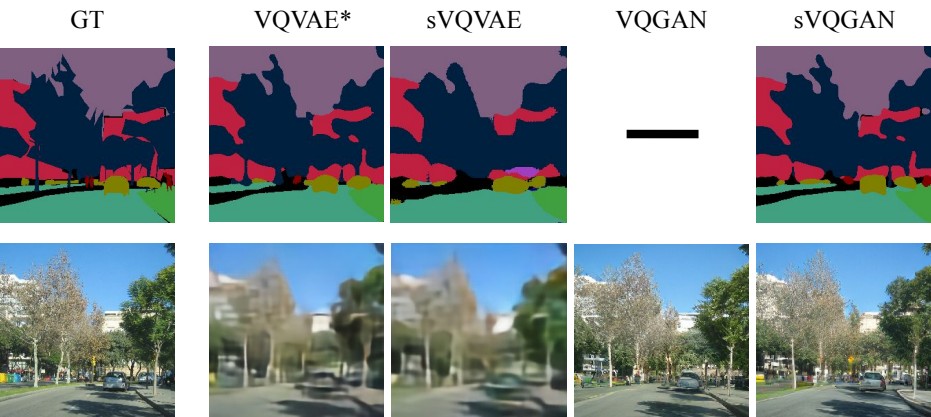

Figure 4: Qualitative results for the auto-encoding of both semantic and image content. *VQVAE is trained individually on each modality, i.e. two different models. Semantic VQ-models (sVQVAE and sVQGAN with $\lambda = 0.1$) are one model each evaluated on both modalities.

purely auto-encoding of each modality individually is desired. This becomes apparent when comparing mIOU as VQVAE improves over sVQVAE by 17.32%, 36.09% and 12.23% for Cityscapes, ADE20k and COCO-Stuff respectively. However, the semantic VQGAN model outperforms all VQ-VAE models when it comes to semantic reconstruction obtaining a mIOU of up to 86.81%, 90.37% and 93.81% when $\lambda = 1.0$. Even for $\lambda = 0.1$ where we weight the image reconstruction more, sVQGAN still outperforms VQVAE only trained on semantic reconstruction by 0.45%, 4.93% and 1.02%. These results indicate, that the VQVAE architecture is not as capable in reconstructing both image and semantics at the same time as VQGAN. Moreover, we can even achieve a better semantic reconstruction when augmenting with image data, compared to when using semantic data alone.

### A.3 QUALITATIVE RESULTS FOR AUTO-ENCODING OF SEMANTIC INFORMATION AND IMAGES WITH VQ-MODELS

In Figure 4 we show qualitative results for auto-encoding of both semantic information and images using different VQ-models. There is a significant difference in image quality between VQVAE and VQGAN models due to the additional discriminator loss. When compressing to a $16 \times 16$ latent space, VQVAE reconstructions exhibit blurriness due to their tendency of rate-distortion trade-offs in their bottleneck as opposed to a rate-perception trade-off (Williams et al., 2020). The outlines of trees and the car are visible, but detail is lost. VQVAE and sVQVAE seem to be on par when it comes to image reconstruction as is also suggested by the quantitative results in Table 2. Moreover, we find that some details in the semantics are lost due to the quantization bottleneck that forces the networks to compress this information, but the general structure is kept intact. For instance, we observe that VQVAE and sVQGAN retain more details in the semantic reconstruction than sVQVAE. The performance of the VQVAE can be explained because semantic reconstruction is the only objective. At the same time, the GAN-based image reconstruction of sVQGAN seems to synergize better with the semantic reconstruction objective than for sVQVAE. For instance, only in sVQGAN, the semantic information about a person on the right border of the image is preserved.

## B ADDITIONAL QUALITATIVE RESULTS

Figure 5 shows more qualitative examples of semantic image synthesis for VQGAN-T and sVQGAN-T. Two examples for each dataset are shown, i.e., COCO-Stuff, ADE20k, Cityscapes. We again observe that semantics are followed more closely for our sVQGAN-T model. In the first row, the goat can only be properly recognized in the generations of sVQGAN-T. In row three, the small lake of water is only present in sVQGAN-T generations, and in row four the plant on the left side of the image is missing for VQGAN-T. Complex city scene also sometimes lack some details

in VQGAN-T generation, e.g., in the fifth row, the car in the center is not synthesized while it is in our sVQGAN-T variant.

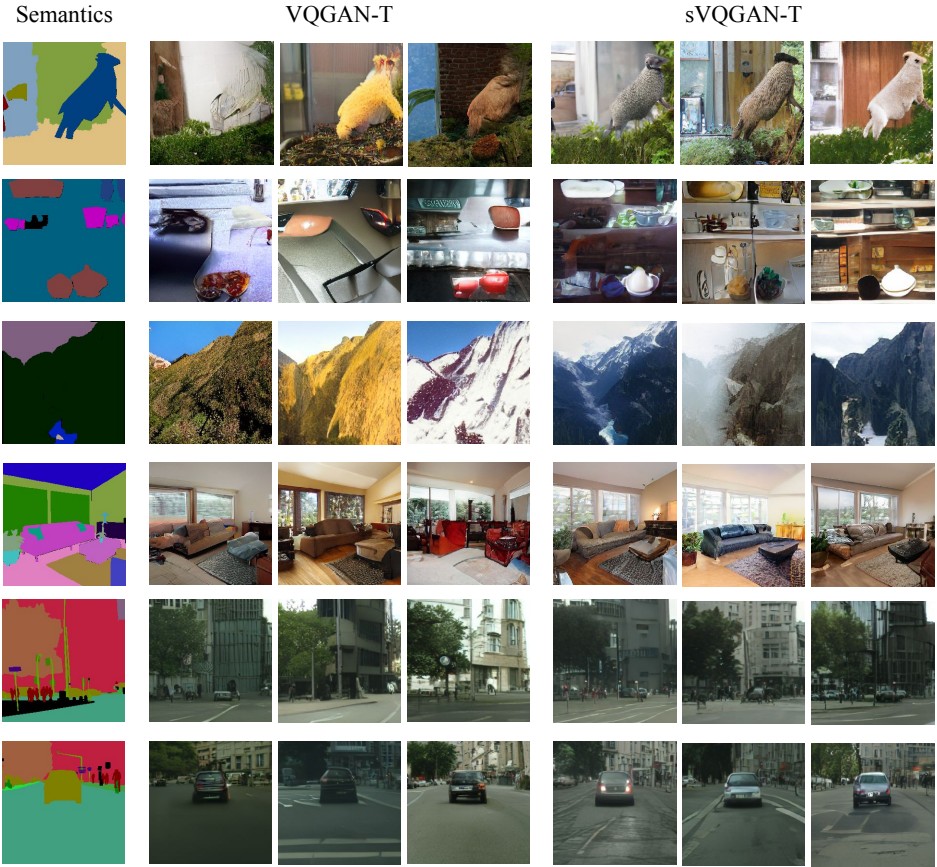

Figure 5: Qualitative results comparing multiple generations of an individually trained VQGAN-T with a semantically coupled sVQGAN-T on COCO-Stuff (top two), ADE20K (middle two) and Cityscapes (bottom two). Semantic details are better retained by sVQGAN-T.

