# OpenReview forum: "Semantic Image Synthesis with Semantically Coupled VQ-Model"
_ICLR.cc/2022/Workshop/DGM4HSD — ICLR 2022 DGM4HSD workshop Poster_

### Official Review · Reviewer_wod4 · 2022-03-23
**Semantic conditioning for VQ generative models with very nice results**

**Rating:** 6
**Confidence:** 4

**Review:**

This work addresses the problem of semantically conditional image synthesis -- that is generating images conditioned on a segmentation map which specifies various objects and background types. This is an important problem in computer graphics.

This work builds on previous vector-quantized image generation methods by training 2 vector quantized models; one for the natural images, and another for the semantic segmentation maps. This creates 2 discrete codebooks -- one for each modality. The work then replaces the standard autoregressvive transformer model for the latent codes with a conditional autoregressive model. This model generates the image-tokens conditioned on the tokens from the segmentation map.

The authors find that this additional conditioning step improves image generation performance scored by the FID, LPIPS, and SSIM (metrics which qualitatively score perceptual image quality).

While I do not find this work to be very theoretically novel, the results are nice and demonstrate the flexibility of the vector-quantized approach to image generation. Therefore, I would advocate for the paper's acceptance.

---

### Official Review · Reviewer_4fLW · 2022-03-28
**The paper is easy to follow with good results**

**Rating:** 7
**Confidence:** 4

**Review:**

**Summary**

This paper proposes an approach for semantic image synthesis. Instead of training the transformer on conditioning latents and image latents separately, the paper proposes to jointly train the model. The proposed approach is able to improve semantic image synthesis on commonly seen semantic image datasets.


**Comments**
1. The paper is well written and easy to follow.
2. The proposed approach achieves strong performance in the experiments. The experiments are detailed.
3. It would be good to provide some theoretical analysis on the performance gain.
4. It would be good to add human evaluations for comparison with the baselines.
5. It would be good to show some failure examples and discuss the limitations.

---

### Decision · Program_Chairs · 2022-03-28

Accept (Poster)